# Universal and divergent P-stereogenic building with camphor-derived 2,3-diols

Yulong Zhang[1], Peichao Zhao [1], Shengnan Sun[1], Qian Wu[1], Enxue Shi [1✉] & Junhua Xiao [1✉]

The access to P-stereogenic motifs has always been considered a very challenging and high attractive mission in modern organic synthesis. While several chiral auxiliaries employed by the practical Jugé-Stephan-like methodology have been developed, new type of readily accessible bifunctional ligands toward P-stereogenic building still remain much desirable. Herein, we present a powerful chiral template, camphor-derived 2,3-diols named CAMDOL, which were designed and synthesized from the commercially cheap camphorquinone in high yields at 50 grams scale with a column-free purification. Diverse P(III)-chiral compounds and their borane forms including phosphinous acids, phosphinites, and phosphines, as well as the corresponding P(V)-chiral compounds including phosphinates, phosphine oxides, phosphi-nothioates, phosphine sulfides, and secondary phosphine oxides were afforded in high yields and $ee$ values through the optimal 2,3-diphenyl CAMDOL platform. An unusual $C^3$-OP bond cleavage following the first P-OC$^2$ bond breaking was observed during the ring-opening process when quenching by NH$_4$Cl solution, which generates a unique but valuable camphor-epoxide scaffold as by-product.

[1] State Key Laboratory of NBC Protection for Civilian, Beijing, China. ✉email: exshi@sina.com; xiao.junhua@pku.edu.cn

Phosphorus asymmetry, matched only by carbon, has been extensively attracting the curiosity of the scientific community especially since the pioneering work by Knowles in the 1960s[1,2]. P-stereogenic compounds feature many peculiar properties, allowing their use in various fields of applications ranging from medicine discovery, material science, to asymmetric catalysis[3–5]. Various P(III)-stereogenic ligands such as the monodentate and bidentate chiral phosphines have proved to be the cornerstone of countless catalytic reactions[5–9], while numerous P(V)-stereogenic compounds such as the new-emerging antisense oligonucleotide (ASO) therapeutics and the COVID-19 pandemic drug Remdesivir display versatile interesting biological properties[10,11]. Therefore, the development of new asymmetric routes to these molecules is crucial to sustain the fast-growing demands for novel chirality-at-phosphorus scaffolds.

For a long time, the asymmetric synthesis of P-chiral compounds has relied heavily on the resolution of racemates until the establishment of the so-called Jugé-Stephan methodology based on the nucleophilic ring-opening of ephedrine-derived borane oxazaphospholidines in a $S_N2@P$ process with organolithium reagents[12]. However, one of the disadvantages of the practical Jugé-Stephan methodology is that ephedrine is a key precursor of the psychotic substance methamphetamine and tightly controlled, thus not amenable for scale-up activities. Therefore, the design of readily accessible and universal chiral auxiliaries with programmable connecting and leaving groups for their installation and detachment on P-atom remains highly desirable and utmostly important in the field of P-asymmetry chemistry.

Indeed, several types of heterobifunctional chiral pools have been developed in the past decades, including the aminoalcohols such as Juge's ephedrine[13], Verdaguer's aminoindanol[14], Andrioletti's 1,2-aminocyclohexanol[15], Framery's glucosamine[16], Kang's pyrrolidinemethanol[17], and Han's 2-(1-aminoethyl)-4-chlorophenol[18], and the thiol-alcohols such as Corey's camphor thiol[19] and the latest Baran's limonene-derived thiol (Fig. 1)[20]. However most of these chiral pools are still suffering from low-availability, high-costage, multi-step synthetic sequences, and scale-up difficulty, and more importantly the effective installation and controllable transformation of the chiral information on the stable P(III) and/or P(V) centers still remain very challenging today.

On the other hand, chiral phosphoric acid (CPA) organocatalysts derived from homobifunctional diols such as BINOL, TADDOL, VAPOL, SPANOL, have been utilized extensively over the last thirty years[21]. However few reliable stereocontrol examples about these diols toward P-stereogenic building have been reported mainly due to that their two hydroxyl groups are actually identical, which makes it too difficult to proceed stereocontrollable transformations. But very recently, synthesis of P-chiral compounds via a kind of axis-to-center strategy by transferring the axial chirality of diols to central chirality located at phosphorus came into feasible. Two typical successful cases involving BINOL have been achieved. In 2020, Murai firstly reported the synthesis of P-chiral phosphonothioates and phosphonates by a two-step alcoholysis of binaphthyl phosphonothioates(Fig. 2b)[22,23]; then in 2021, Feringa reported the synthesis of P-chiral phosphine oxides by Pd-catalyzed arylation of phosphoramidites in presence of $Cs_2CO_3$ (Fig. 2a)[24]. Unfortunately, the above approaches are not enough efficient and compatible such as demanding acid-catalyzed methanolysis to pre-cleavage the unreactive P–N bond, or unsatisfactory stereoselectivity and very limited scope with only few examples. Moreover, the main drawback of the current diol-based strategy is that they are not amenable for the synthesis of structure e-diverse P(III)-chiral compounds and the conversions into the P(V)-molecules. Therefore, the development of a new type of diol-type template reagents capable of achieving both introduction and conversion of the P(III)- and/or P(V)-stereogenic centers with robust stereoselectivity remains high valuable in present.

When opting for an inventive chiral pool fulfilled the above requirements, we recently focused on the camphor-related diols which are rarely interested and explored before in organophosphorus synthesis. As one of the nature's privileged scaffolds readily available in both enantiomeric forms, camphor undergoes a wide variety of chemical transformations, thus enabling the preparation of structurally and functionally diverse products[25]. However, the vast majority of the camphor-derived catalysts in present are frameworks connected to a chiral functionality such as pyrrolidine, taking the camphor skeleton only as the subsidiary part. But a recent interesting report by Huseyinov[26] about the application of organophosphorus acid from camphor 2,3-dimethyl-2,3-diol in Hantzsch reaction inspired us to develop a kind of new chiral pool of camphor-type diols. In theory, the commercially available inexpensive camphorquinone can react with two equivalent organolithiums to afford the corresponding camphor 2,3-diols, thus enough hindrances may be achieved if introducing two bulky endo 2,3-disubstituted groups onto the camphor backbone. To our best knowledge, no such camphor 2,3-diols were full investigated and used by the axis-to-center methodology before. Herein, we present an exceptional camphor 2,3-diols template to realize divergent stereoselective synthesis of phosphinous acid boranes which can be subsequently converted into the various P(III)- and P(V)-chiral compounds via known methods (Fig. 2c).

## Results and discussion

### Synthesis of camphor-derived 2,3-diols (CAMDOL). Camphorquinone, as a kind of diketones usually used as

**Fig. 1 Representative heterobifunctional auxiliaries employed in P-asymmetry by Jugé-Stephan type transformations. a** aminoalcohol-type auxiliaries. **b** thiol-alcohol-type auxiliaries.

**Previous works:**

**(a)** Feringa's, *Nat. Cat.*, 2021

**(b)** Murai's, *J. Org. Chem.*, 2020

**This work:**

**(c)**

- **CAMDOL ligand:**
  - air and moisture stable
  - 50-gram-level scale-up
  - column-free isolation
  - up to 88% recovery
  - low-cost materials
- **P*-Building:**
  - P(III) & P(V)-compatible
  - P-enantiodivergent

**49 examples**
**up to: 98% yield;**
**99% ee**

**Fig. 2 P-stereogenic building based on bifunctional diol-type chiral auxiliaries. a** Feringa's synthesis of P-chiral phosphine oxides by Pd-catalyzed arylation of BINOL-derived phosphoramidites. **b** Murai's synthesis of P-chiral phosphonothioates by two-step alcoholysis of BINOL-derived phosphonothioates. **c** CAMDOL-enabled synthesis of diverse P(III)- and P(V)-chiral compounds by us.

photoinitiator, featuring a rigid but nonaxisymmetry, is commercially available in both (+)- and (-)-isomers with quite low prices. We firstly took the (-)-camphorquinone as the starting material to test our idea. As expected, (-)-camphorquinone was found to react smoothly with Grignard reagents or organolithium reagents via an *endo*-specifically nucleophilic addition, giving the target 2,3-substituted diols including dimethyl, diethyl, dibutyl, dibenzyl, and diphenyl 2,3-diols (Fig. 3). Notably, the camphor-derived 2,3-diols, herein named as CAMDOL, which would generally precipitate as white solids from the concentrated organic solvent after 2 h refluxing in THF following with saturated NH$_4$Cl quenching. The high-pure products of CAMDOL **1a-e**, were thus obtained only by simple filtration without any further column chromatography isolation. Pleasingly, all of the carmphor-diol compounds were found air and moisture stable enough with experimental handling. Moreover, the reaction showed outstanding scalability that the 2,3-diphenyl CAMDOL **1e**, supposed to be the most bulky ligand, could be easily prepared in 85% yield even at a high to 50-gram-level operation. According to the X-ray analysis of **1e**, significant hydrogen-bond interaction between the two hydroxyl groups and some torsion of two phenyl groups are formed, demonstrating its rigid space orientation (Fig. 3). But when we tried to install more bulky groups such as

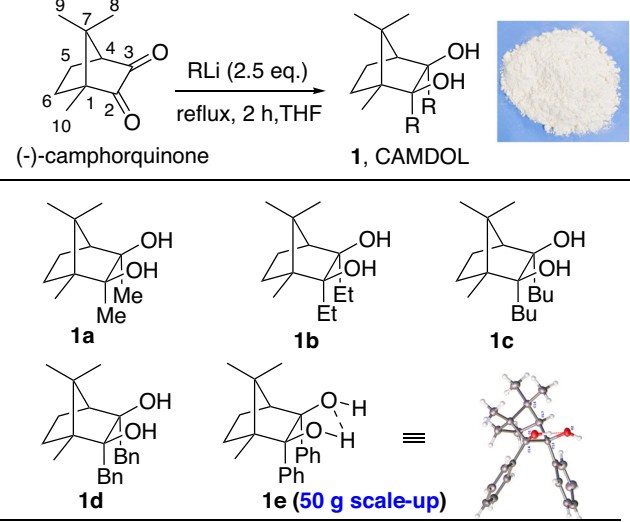

**Fig. 3 Column-free and scale-up synthesis of CAMDOL.** Reaction conditions: see Methods in this main manuscript text. X-ray spectra: see Supplementary Data 2.

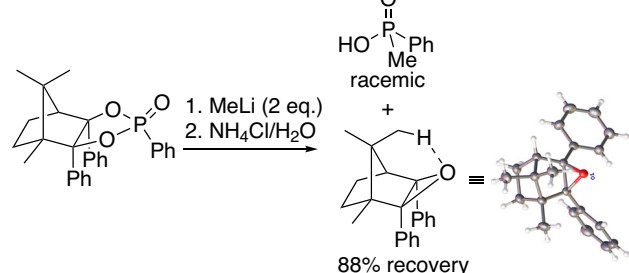

**Fig. 4 Phosphorylation and phosphitylation of CAMDOL 1e.** Reaction conditions: **1e** (1.0 eq.), RPCl₂ (1.5 eq.), Et₃N (2.5 eq.), in THF at 0 °C for 2 h; then added conc. H₂O₂ at rt for 30 min. **3a, 2a–c** not isolated.

**Fig. 5 Ring-opening reaction of P*-phosphonate 3c.** Experiment details: see Supplementary Information—Initial explorations with CAMDOL-phosphonates. X-ray spectra: see Supplementary Data 3.

iso-propyl, xenyl, p-tBu-phenyl, 2-naphthyl onto the camphor skeleton, the reactions were found halted at the once-addition stage that only the less shielding 3-position C=O was alkylated even with excess organolithium reagents at refluxing temperature.

**CAMDOL-enabled enantioselective synthesis of phosphinous acid boranes.** With the chiral pool of CAMDOL in hand, we then investigated their feasibility in P-stereogenic building. Considering that CAMDOL **1e**, possessing the highest steric effect, so we firstly examined the phosphorylation of **1e** by POCl₃ and phosphitylation of **1e** by PCl₃. Very surprisingly, no reaction between **1e** and POCl₃ was observed even after refluxing 12 h in THF, whereas when stirring **1e** with PCl₃ by 30 min at 0 °C in presence of Et₃N, the backbone of **1e** was then found completely anchored to P(III)-atom forming a chloro dioxaphospholidine motif **2a** as desired. Compared with POCl₃, PCl₃ is relatively more reactive and less bulky, which may contribute to the above distinguished behaviour when introducing P-atom onto the CAMDOL's tertiary hydroxyl groups, similar to that of TADDOL's reactions[27].

The commercially available MePCl₂ and PhPCl₂ gave **2b** and **2c** under the similar conditions. Encouragingly, the directly unavailable P(V) phosphorochloridates and phosphonates could be obtained effectively from the crude P(III)-intermediates by H₂O₂ oxidation in an one-pot procedure. By this manner, P(V)-compounds **3b-c** were synthesized from CAMDOL **1e**, while the unstable P(III)-compounds **2a-c** and P(V)-compound **3a** were generally used as crude in the following reactions (Fig. 4).

Next, we took the P*-synthon **3c** to test the ring opening reaction with typical organometallic reagents. To our surprise, the second P–O bond cleavage never took place even with excess of MeLi at refluxing temperature, although the first P-O bond cleavage occurred very quickly. However, when quenching the mixture by saturated NH₄Cl solution, a racemic PhMeP(O)OH accompanied with an unanticipated camphor-epoxide were isolated (Fig. 5), which indicates that the reaction probably underwent with an unusual intramolecular cyclization process similar to the Baran's Π-reagent[28]. The detaching order of the two P–O bonds will be discussed in the following proposed reaction mechanism.

It should be pointed out that the camphor-epoxide would always crystallize out from the concentrated aqueous mixture at the end, and could be recovered in 88% yield. The unique by-product camphor-epoxide was confirmed by NMR spectroscopy, high-resolution mass spectrometry (HRMS) and single crystal X-ray analysis. The X-ray spectra demonstrates that a clear hydrogen-bond interaction forms between the epoxide oxygen and C-8 hydrogen in the epoxide scaffold.

Obviously, the above experiment suggests that it's not accessible to achieve P-chirality since the two oxygen atoms of

P=O and P–OH in the product are not discriminal. But looking from another perspective, we speculated that diastereoselectivity should be realized if introducing an unequivalent atom, such as borane or sulfur, to replace the oxygen of P=O group. The much more attractive P(III)-BH₃ were preferred, as surrogates of air-sensitive P(III) compounds, mainly due to their easy handling, purification and transformation[29–32]. More importantly, the P(III) phosphinous acids, products after deprotection of BH₃, are able to tautomerise to the P(V) secondary phosphine oxides (SPOs) which are high valuable compounds with a huge potential in asymmetric synthesis, catalysis and coordination chemistry[33–36]. The advantages of new-emerging P-chiral SPOs over the most often used phosphine ligands are threefold: air and moisture stability, supramolecular bidentate formation, and bifunctional ligands activity. However, only a few efficient examples are known so far. The most practical approach employed for SPOs was a multi-step procedure starting from the diastereomerically enriched menthyl H-phosphinate precursors, which however normally need twice recrystallization manipulations at low temperature, unfortunately always in very low yields[37,38]. More general and competitive enantioselective methodology toward SPOs still remain a very challenging topic now.

Inspired by the above idea, we then examined the Jugé-Stephan type reaction of the camphor-derived P*-phospholidine boranes **4ca-ce** prepared conveniently by BH₃/SMe₂ in high yields. As shown in Table 1, **4ce** could be converted into **5aa** in 70% ee at room temperature by the smallest Grignard reagent MeMgBr (1 mmol/L) in THF (Table 1, Entry 5), whereas **4ca-cd** gave relatively lower enantioselectivities (Table 1, Entries 1–4). Better results could be obtained at lower temperatures of −40 °C (Table 1, Entries 6–8), but leading to prolonged reaction time and declined yield (Table 1, Entry 9). Screening of solvents demonstrated that toluene seemed to be the best choice (Table 1, Entries 10–15), giving **5aa** in 86% yield and 90% ee (Table 1, Entry 13). Encouragingly, when improving the mixture concentration from 0.067 mmol/L to 0.133 mmol/L by using 2.5 eq. MeMgBr and half toluene solvent, the reaction proceeded with 92% ee (Table 1, Entry 16). Further reducing MeMgBr to 1.5 eq., up to 97% ee was afforded with 86% isolated yield of **5aa** (Table 1, Entry 17). Nevertheless, 1.1 eq. MeMgBr would give slightly diminished yield and ee value as well as longer reaction time (Table 1, Entry 18).

Having established the optimized conditions (Table 1, Entry 17), the scope of P-chiral phosphinous acid boranes **5a** was explored (Fig. 6). A wide array of primary alkyl, cycloalkyl, and alkenyl Grignard reagents were well tolerated in this process, providing the corresponding products **5aa-5ah** in 80–92% yields and 90–97% ee values. But the alkynyl and aryl Grignard reagents only gave products **5ai-5ak** with moderate yields and

**Table 1 Scope of the CAMDOL-enabled preparation of 5aaᵃ.**

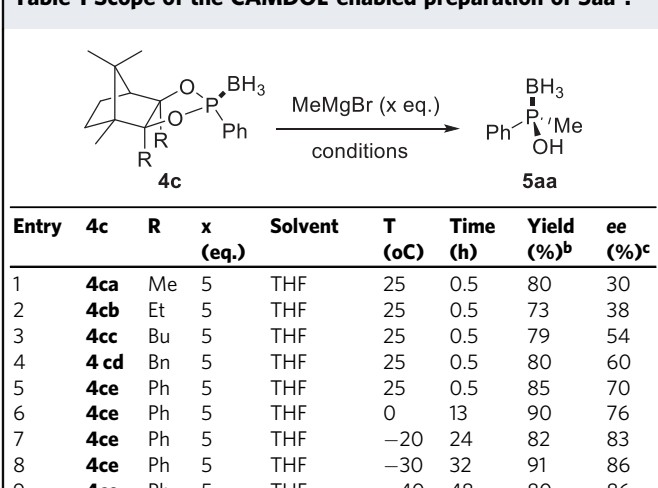

| Entry | 4c | R | x (eq.) | Solvent | T (°C) | Time (h) | Yield (%)ᵇ | ee (%)ᶜ |
|---|---|---|---|---|---|---|---|---|
| 1 | **4ca** | Me | 5 | THF | 25 | 0.5 | 80 | 30 |
| 2 | **4cb** | Et | 5 | THF | 25 | 0.5 | 73 | 38 |
| 3 | **4cc** | Bu | 5 | THF | 25 | 0.5 | 79 | 54 |
| 4 | **4 cd** | Bn | 5 | THF | 25 | 0.5 | 80 | 60 |
| 5 | **4ce** | Ph | 5 | THF | 25 | 0.5 | 85 | 70 |
| 6 | **4ce** | Ph | 5 | THF | 0 | 13 | 90 | 76 |
| 7 | **4ce** | Ph | 5 | THF | −20 | 24 | 82 | 83 |
| 8 | **4ce** | Ph | 5 | THF | −30 | 32 | 91 | 86 |
| 9 | **4ce** | Ph | 5 | THF | −40 | 48 | 80 | 86 |
| 10 | **4ce** | Ph | 5 | DCM | −30 | 36 | 88 | 84 |
| 11 | **4ce** | Ph | 5 | Et₂O | −30 | 36 | 89 | 70 |
| 12 | **4ce** | Ph | 5 | CCl₄ | −30 | 36 | 80 | 76 |
| 13 | **4ce** | Ph | 5 | Toluene | −30 | 36 | 86 | 90 |
| 14 | **4ce** | Ph | 5 | *m*-Xylene | −30 | 36 | 85 | 85 |
| 15 | **4ce** | Ph | 5 | Meitylene | −30 | 36 | 88 | 84 |
| 16ᵈ | **4ce** | Ph | 2.5 | Toluene | −30 | 36 | 86 | 92 |
| 17ᵈ | **4ce** | Ph | 1.5 | Toluene | −30 | 36 | 86 | 97 |
| 18ᵈ | **4ce** | Ph | 1.1 | Toluene | −30 | 48 | 85 | 95 |

[a] Reaction conditions: **4c** (0.5 mmol), MeMgBr (x/2 mmol), solvent (5 mL); [b] Isolated yields; [c] Determined by HPLC; [d] **4ce** (0.5 mmol), MeMgBr (x/2 mmol), solvent (2.5 mL).

**Fig. 6 Preparation of 5a via Grignard reagents.** Reaction conditions: **4ce** (1.0 eq.), RMgBr (1.5 eq.), in PhMe at −30 °C for 36 h; then NH₄Cl solution at rt.

enantioselectivities. More bulky secondary and tertiary alkyl, as well as *ortho*-substituted aryl Grignard reagents were all found not applicable in this way, possibly due to the low reactivity of magnesium reagents compared with lithium reagents, which was proved in our following test.

To our delight, excellent yields and stereoselectivities were both performed as expected when employing the organolithium reagents in place of organomagnesium reagents (Fig. 7). Both

bulky secondary and tertiary alkyl lithium proceeded smoothly and gave the wanted products **5ba** and **5bb** in 97% and 93% *ee* respectively, which were unavailable by the Grignard reagents. Aryl lithium with electron-donating or electron-withdrawing substituents at different positions were all compatible, affording products **5bc-5bk** with 95% to 99% *ee* values. The condensed and hetero aromatic lithium reagents proceeded almost enantiospecifically, giving the products **5bl, 5bm, 5bn,** and **5bo** with 99% *ee* values. The absolute configuration of **5bo** was further determined by X-ray diffraction analysis.

The reaction between the lest bulky methyl dioxaphospholidine borane **4be** and *ortho*-methoxylphenyl lithium also gained product **5 bp** with 97% *ee* value, but for the less-bulky *meta/para*-methoxyl phenyl lithium reagents, lower enantioselectivities of 82% and 65% *ee* were obtained for products **5bq** and **5br**. The distinct differences of steric effects between phenyl and methyl groups on the P-center, as well as the steric shielding from different positions (*ortho*, *meta*, and *para*) at the aromatic ring, should contribute mainly to the above enantioselective alterations.

With respect to that most of the alkyl/aryl phosphonodichlorides (RPCl₂) are not commercially available, we then turned to further explore the universality and practicability of CAMDOL by using 2-chloro 1,3,2-dioxaphospholidine borane **4ae** as the starting material. Some bulky groups which are not included in Fig. 7 but potentially emerged in P-asymmetry motifs were selected. As shown in Fig. 8, phosphinous acid boranes compound **5ca** with ethenyl and biphenyl, compound **5cb** with cyclohexyl and naphthyl, and compound **5cc** with 4-fluorophenyl and 2-ethenylphenyl attached on P-atom were thus obtained in 96%, 97%, and 98% *ee* values respectively.

It is interesting to note that, stereodivergent preparation of both enantiomers of the P-stereogenic molecules ($R_P$)-**5bo** and ($S_P$)-**5bo** were easily achieved in very close yields and the same 99% *ee* values with CAMDOL only by using the (+)- or (-)-camphorquinone enantiomeric forms as the starting materials respectively (Fig. 9).

Alternatively, according to the Jugé's strategy[12], the stereoselective formation and flexibility of P-stereogenic centre were also easily afforded simply by switching the sequence of organomagnesium additions, providing either ($S_P$)-**5aa** or ($R_P$)-**5aa** with equally minimal effort (Fig. 10).

**Stereoretentive P\*-transformations of phosphinic acid boranes.** With enantiopure phosphinic acid boranes in hand, our search for their derivation reactions toward synthesis of notable P(III)-chiral ligands commenced. After 10 mmol scale preparation, compound **5bf** was firstly converted into PAMP-BH₃ via *O*-methylation[39,40] by TMSCHN₂ (firstly attempt by MeI but failed) and then nucleophilic substitution by MeLi. The following deprotection of BH₃ by DABCO afforded the well-known P(III)-ligand PAMP in 74% total yield without compromising distinct stereochemical integrity (Fig. 11).

Then, we carried out exploration of the scope toward various kinds of P(III)- and P(V)-chiral compounds. Figure 12 illustrates the scope and sequence of CAMDOL-enabled synthesis of some selected P-chirogenic compounds. The *O*-methylation of phosphinous acid boranes **5bi** and **5bo** with TMSCHN₂ gave the corresponding homochiral phosphinite boranes intermediates **6a-b** in quantitative yields as an excellent starting point for the next preparation of diverse P-chiral molecules. Compounds **6a-b** were able to be transformed into phosphinates **7a-b** and phosphinothioates **8a-b** respectively with 98–99% P-chirality retention when treated by DABCO and then *m*-CPBA, or by DABCO and then elemental sulfur at 50 °C in THF. Compound **6b** could also

The products shown in Fig. 6:

**5aa** 86%; 97% ee
**5ab** 84%; 94% ee
**5ac** 92%; 90% ee
**5ad** 90%; 91% ee
**5ae** 80%; 93% ee
**5af** 92%; 95% ee
**5ag** 92%; 91% ee
**5ah** 84%; 95% ee
**5ai** 82%; 80% ee
**5aj** 58%; 74% ee
**5ak** 80%; 83% ee

**Fig. 7 Preparation of 5b via organolithium reagents.** Reaction conditions: **4ce/4be** (1.0 eq.), RLi (1.5 eq.), in THF at −78 °C for 12 h; then NH$_4$Cl solution at rt. X-ray spectra: see Supplementary Data 4.

be converted into phosphine **9b** with 96% P-chirality inversion when treated by lithium reagents at −30 °C and then DABCO at 50 °C in THF. Compound **9b** then was transformed into phosphine oxide **10b** or phosphine sulfide **11b** respectively without any P-chirality loss under the same conditions. All the above reactions gave very high to quantitative yields of the desired products.

Moreover, the deboronation reaction of phosphinous acid boranes was also examined by the known HBF$_4$-mediated method[41] to afford the P-chiral SPOs. Compounds **12a-d** were thus successfully synthesized with 95-98% enantioselectivities reserved (Fig. 13). Compared with the conventional cumbersome menthol-based methodology, our CAMDOL-enabled access to SPOs possess several extinct advantages such as higher overall output, better substrates scopes and easier experimental handling, thus can be potentially used as a general protocol for synthesis of diverse SPOs in the further.

**Plausible reaction mechanism**. It's undoubtable that two *pseudo*-homobifunctional OH groups in CAMDOL are not really sterically identical, especially in the ring-opening process. However, the only difference between them is the C$^{10}$-methyl group. Therefore, based on the above reaction results observed, the CAMDOL-involved Jugé-Stephan-like transformation possibly proceeds in the following way. The organomental reagent prefers to attack the frontside C$^2$-hydroxyl P-O bond rather than the backside C$^3$-hydroxyl P-O bond probably due to the C$^{10}$-hydrogen bond chelating with RM to form a kind of stabilized six-membered ring. Under the mild acidic conditions with NH$_4$Cl solution, the C$^3$-OP bond undergoes a heterolytic cleavage leading to the formation of P-chirality retented phosphinous acid product and the stabilized tertiary benzylic carbocation **Int_CAM**. The following intramolecular S$_N$1 nucleophilic attack of C$^2$-hydroxyl group at C$^3$-carbocation of **Int_CAM** then leads to the formation of camphor-epoxide (Fig. 14).

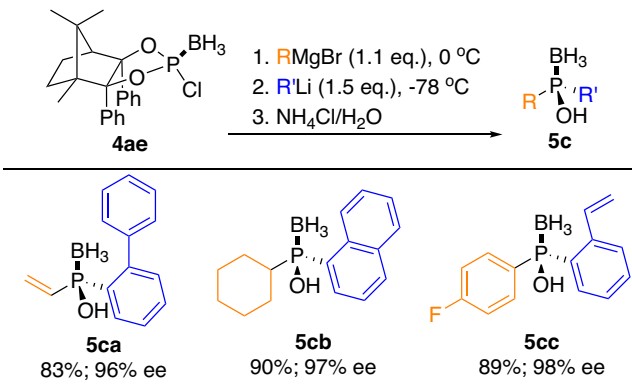

**Fig. 8 Preparation of 5c from chloro dioxaphospholidine 4ae.** Reaction conditions: **4ae** (1.0 eq.), RMgBr (1.1 eq.), in THF at 0 °C for 1 h; then R'Li (1.5 eq.), at −78 °C for 12 h; then NH₄Cl solution at rt.

**Fig. 9 Enantiodivergent synthesis of 5bo by using different camphorquinone enantiomeric forms.** Reaction conditions: **4ce** (1.0 eq.), RLi (1.5 eq.), in THF at −78 °C for 12 h; then NH₄Cl solution at rt.

**Fig. 10 Enantiodivergent synthesis of 5aa by switching the organometallic reagents addition sequence.** Reaction conditions: **4ae** (1.0 eq.), RMgBr (1.1 eq.), in THF at 0 °C for 1 h; then R'MgBr (1.5 eq.), at −30 °C for 36 h; then NH₄Cl solution at rt.

## Conclusion and outlook

In summary, a type of camphor-based chiral pool named CAMDOL, featuring salient stability, excellent scalability, and high recoverability, was designed and developed from the commercially inexpensive camphorquinone. This powerful C¹-asymmetric diol-type chiral auxiliary provides broad admission to otherwise challenging sectors of various P-stereogenic synthesis with impressive stereocontrol performances. The optimal 2,3-

**Fig. 11 Preparation of P(III)-ligand PAMP via 5bf.** Experiment details: see Supplementary Information—General procedure for synthesis of P(III)-ligand PAMP.

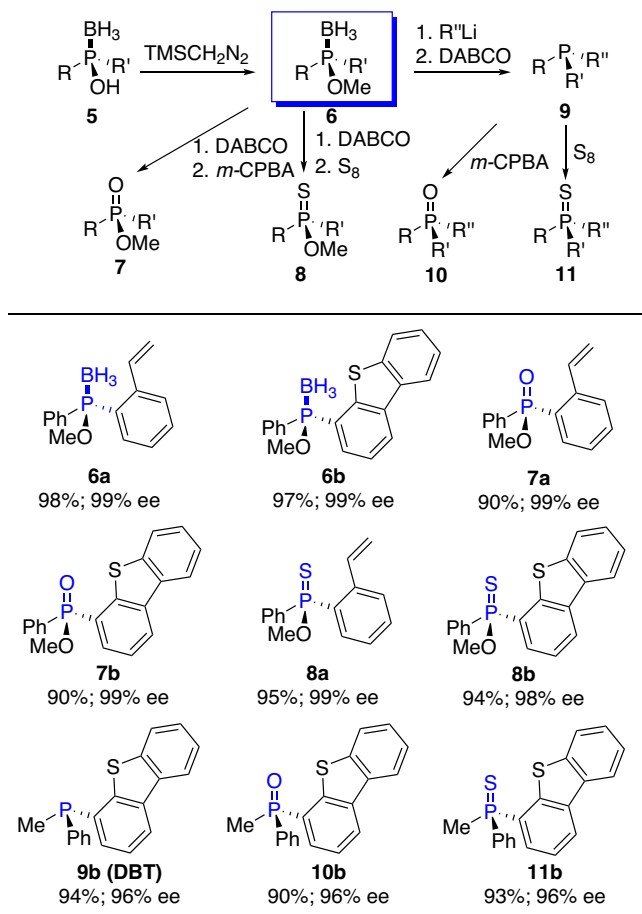

**Fig. 12 CAMDOL-enabled asymmetric synthesis of diverse P(III) & P(V)-stereogenic compounds 6-11.** Experiment details: see Supplementary Information (P55-74).

diphenyl CAMDOL enabled the diastereoselective synthesis of P(III)-chiral phosphonious acid boranes which as universal synthons can be further converted into a broad array of structurally diverse P(III)- and P(V)-chiral compounds. The well-known phosphine ligands PAMP and DBT[42] (**9b**), as well as the popular ambidente SPO ligands, were all afforded in high enantioselectivities with remarkable operational simplicity. Stereodivergent preparations of both enantiomers of the P-chirogenic

**Fig. 13 General synthesis of P-chiral SPOs 12 from 5.** Reaction conditions: **5** (1.0 eq.), HBF$_4$ (2.0 eq.), in CH$_2$Cl$_2$ at 0 °C for 0.5 h; then NaHCO$_3$ solution at rt.

**Fig. 14 Proposed CAMDOL-involved reaction mechanism.** Plausible mechanism of ring-opening process of CAMDOL-derived phosphinous acid boranes to illustrate the P-chiral center transformation.

molecules were also easily achieved either by using the different camphorquinone enantiomeric forms or by switching the organometallic reagents addition sequence.

More importantly, the full capability of CAMDOL-template for various asymmetric organophosphorus chemistry, such as the P(III)-ligands synthesis, phosphor(n)othioates preparation, α-carbon functionalization, phospha-Aldol addition, as well as the CPA catalysts application, are yet to be in-depth revealed, which we deem much attractive objectives to be beyond the scope of this article, and will soon be presented in our following works.

## Methods
For instrumentation and materials, see Supplementary Methods—General Experimental.

**General procedure for synthesis of CAMDOL**. To a flamed-dried 250 mL round-bottom flask was charged (-)-camphorquinone (1 eq.). The flask was evacuated and back filled with argon, after which anhydrous THF (100 mL, 0.1 M) was introduced via a cannula. Organolithium reagent (2.5 eq.) was then added dropwise and the resulting solution was allowed to stir at refluxing temperature until TLC showed complete consumption of starting material. The solution was allowed to come back to room temperature and then quenched with slow addition of saturated aqueous NH$_4$Cl solution (40 mL) and diluted with water (80 mL) and EtOAc (150 mL). The two layers were separated, and the aqueous layer was washed twice with EtOAc (2 × 80 mL). The combined organic layers were washed with saturated aqueous brine (50 mL), and dried over anhydrous Na$_2$SO$_4$. The mixture was then filtered, concentrated in vacuo, and purified by simple filtration to afford the desired product.

**General procedure for synthesis of phosphinous acid-boranes 5b**. The organolithium reagent (1.5 mmol, 1.5 eq.) in 14 mL of THF was cooled to −78 °C if not already at such temperature. A solution of the starting material **4** (1.0 mmol, 1.0 eq.) in 6 mL THF was prepared in a flame-dried flask under argon atmosphere, which was then added dropwise to the flask containing the organolithium reagent. The resulting mixture was stirred for 12 h while being kept at −78 °C. After $^{31}$P NMR analysis of a small aliquot showed complete consumption of starting material, the reaction was warmed to room temperature. To the resulting mixture was added saturated aqueous NH$_4$Cl solution (20 mL) and EtOAc (40 mL). The

layers were separated, and the aqueous layer was washed with EtOAc (2 × 20 mL). The combined organic layers were washed with brine (20 mL), dried over anhydrous Na$_2$SO$_4$, filtered and concentrated. The residue was purified by silica gel chromatography to afford the desired product.

**Spectroscopic and X-ray data of products**. See Supplementary Data 1–4.

## Data availability
The authors declare that the data supporting the findings of this study are available within the article and Supplementary Information. For experimental details and compounds characterization, see Supplementary Information. For NMR spectra, see Supplementary Data 1. For X-ray crystallography, see Supplementary Data 2–4 or obtained free of charge from The Cambridge Crystallographic Data Centre with the accession codes CDCC #2268813 (**1e**), #2233676 (camphor-epoxide), #2251985 (**5bo**) via www.ccdc.cam.ac.uk/data_request/cif.

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

## Author contributions

Y.Z. optimized P-stereogenic building methodology, performed synthetic experiments, characterized compounds, and prepared the manuscript. P.Z. performed CAMDOL's synthetic experiments. S.S. and Q.W. performed stereodivergent preparation experiments. E.S. and J.X. directed the whole study and involved in all aspects of the experimental design, data analysis, and manuscript preparation.

## Competing interests

The authors declare no competing interest.
