## [Peer Review File · Communications Chemistry]

Reviewers' comments:

Reviewer #1 (Remarks to the Author):

Chemistry of chiral organophosphorus compound is of utmost importance in the field of chemical sciences. In organic synthesis new ligands based on chiral phosphines are routinely used to facilitate challenging asymmetric transformations. Moreover, in the field of medicinal chemistry chiral phosphate functionalities play critical role in development of new oligonucleotide therapeutics. Therefore, development of new asymmetric routes to these molecules is crucial to sustain growing demand for novel chiral-at-phosphorus scaffolds. In recent years, numerous practical solutions to this problem were presented either based on chiral auxiliaries (ACS Cent. Sci. 2021, 1473) or catalytic approach (Nat. Chem. 2023, 714).

In this article authors introduce new type of camphorquinone-derived chiral auxiliary (CAMDOL), used for the stereocontrolled synthesis of chiral P(III) and P(V) compounds. The authors describe in detail optimization efforts leading to the development of the reagent and its later application in synthesis of diverse phosphinous acid boranes, as well as their further transformations leading to a range of P(III) and P(V)-based molecules. The products are obtained in good to excellent yields and with high stereocontrol, by treating the chiral-P(III) precursor 4c with organometallic reagents (Grignard or Li-based). The major novelty of this work is the application of new chiral-pool-derived auxiliary to the synthesis of asymmetric P(III) reagents. In my opinion this work could be of use in development of new type of chiral phosphine ligands and phosphinous acid boranes. On the other hand, applications of this methodology to access more complex P(V) scaffolds (i.e. nucleotide isosteres) will likely be limited due to the use of highly reactive and moisture-sensitive P(III)-reagents (i.e. RPCl_2).

Overall, I believe that that this manuscript meets the level of novelty required for the publication in Communications Chemistry.

Bellow I enlisted some comments and corrections to be addressed by the authors before final publication:

1. The proposed intramolecular ring closing of epoxide (accompanied by elimination of the phosphinate) is unlikely to proceed via $\text{S}_{\text{N}}2$ -like mechanism as depicted on Fig.11 (Int_CAM). Formation of the epoxide on a 6-membered ring via $\text{S}_{\text{N}}2$ requires antiperiplanar orientation of nucleophilic O and leaving group, which in case of Int_CAM is not possible. It is more likely that this step follows $\text{S}_{\text{N}}1$ mechanism, i.e. under mildly acidic conditions (NH_4Cl) C-OP bond undergoes heterolytic cleavage leading to phosphinate anion and stabilized tertiary benzylic carbocation, followed by the nucleophilic attack of hydroxyl group leading to epoxide.
2. Since one of the major points of the publication is synthesis of chiral phosphinous acid boranes, the authors should cite state-of-the-art publications describing strategies for the asymmetric synthesis of these molecules (e.g. menthol-auxiliary-based approach developed by Buono et al: Chem. Commun. 2008, 3031 doi: 10.1039/B802817F; JACS 2011, 10728 doi: 10.1021/ja2034816).
3. The authors state in the text that both enantiomers of the product can be obtained through selection of desired CAMDOL auxiliary (i.e. derived from (+)- or (-)-camphorquinone). However, all compounds presented in the paper are obtained from (-)-camphorquinone-derived CAMDOL. The authors should back up their statement by comparing yield and ee for enantiomers of one of the products derived from both enantiomers of CAMDOL.
4. In general, there are a lot of grammatical and stylistic errors throughout the text. Before publication I

recommend that the manuscript should be partially rewritten with a help of native English speaker.

5. Supporting Information: Multiplicity of P signal in ^{31}P NMR for multiple phosphinous acid boranes (compounds 4ce, 4be, 5ai, 5aj, 5bd, 5bh, 5bi, 5bj, 5bk, 5bl, 5bm, 5bn, 5bo, 5cb, 6a, 6b) is incorrectly assigned as doublet. ^{10}B nucleus has spin = $3/2$, which gives rise to 1:1:1:1 quartet on ^{31}P NMR. Apparent doublet is results from the signal broadening.

6. Supporting Information: ^{13}C NMR spectrum for compound 1b is missing.

Reviewer #2 (Remarks to the Author):

The manuscript of Shi, Xiao and coworkers describes the use of readily available camphor-derived 2,3-diols (CAMDOLs) as chiral auxiliaries for the preparation of optically pure, P-stereogenic P(III) and P(V) derivatives, including phosphine-boranes, phosphinous acid-boranes, phosphinites, phosphine oxides and phosphinates, among others.

The manuscript describes a new family of chiral, C₁-symmetric diols, derived from camphorquinone as heterobifunctional auxiliaries for the preparation of optically pure P-stereogenic compounds by successive substitutions at phosphorus. The key compound and the most studied is the dioxaphospholidine-borane 4ce.

Despite the long history and the activity in the field of P-stereogenic ligands in the last 25 years, especially as ligands for enantioselective catalysis, there are still very few relatively versatile synthetic methodologies. Indeed, only the Jugé-Stephan method (based on ephedrine) and the Evans-Imamoto method (based on sparteine) are relatively general.

The idea behind the manuscript –using a heterobifunctional auxiliary for the preparation of P-stereogenic compounds by successive substitutions– has been known for a long time as Figure 1 depicts. In this figure, however, only the Jugé's ephedrine method is commonly used.

In this context, the use of a novel cheap and available auxiliary and the large number of compounds described in the paper is very interesting and for this reason I am quite supportive about the acceptance of the manuscript.

Despite this, I have some comments in order to improve the results presented in the manuscript.

- The Jugé-Stephan method. The Jugé-Stephan Method (reference 5) is based on ephedrine but in several occasions in the paper (for example in the caption of Figure 1) it seems to imply that the method uses any bifunctional auxiliary. This is not correct and has to be corrected to make it clear that the Jugé-Stephan methodology is based on ephedrine but that other methods are similar, but using other heterobifunctional chiral auxiliaries.
- Advantages of CAMDOLs. One of the disadvantages of the Jugé-Stephan methodology is that ephedrine is a key precursor of the psychotic substance methamphetamine and therefore is tightly controlled, this could be stressed in the paper as an additional advantage of the methodology presented.

- Phosphinous acid-boranes. The methodology described in the manuscript yields phosphinous-acid boranes, an understudied type of auxiliaries that is used to prepare phosphinite-boranes, phosphine-boranes, oxides and sulfides. This is perfectly fine but I think it could be improved in two ways:

- o The conversion of phosphinous acid-boranes to phosphinite-boranes by O-methylation with trimethylsilyldiazomethane is not so common in the literature and according to the manuscript appears to be quite general. This reaction should be emphasized and referenced appropriately in the manuscript.
- o The deboronation of phosphinous acid-boranes presented in the manuscript would furnish Secondary Phosphine Oxides (SPOs), which are very interesting compounds since they tautomerise to the P(III) phosphinous-acid, which are competent ligands with a huge potential in catalysis. Therefore, the deprotection of a few phosphinous acid-boranes 5 should be described and references about SPOs (for instance *Catal. Sci. Technol.*, 2019, 9, 5504-5561; *ChemcatChem*, 2020, 12, 3982-3994; *Synthesis*, 2022, 54, 271-280) added to the manuscript.

Overall, I support the publication to *Communications Chemistry* if the points raised are appropriately addressed.

Reviewer #3 (Remarks to the Author):

The manuscript by Shi and Xiao describes the synthesis of P-stereogenic P(III) and P(V) compounds using a camphor-based chiral auxiliary as a template for installing substituents at phosphorus. While I find the manuscript and Supporting Information well prepared, I do not find the novelty sufficient for a high impact journal such as *Communications Chemistry*. The methodology is similar to the ephedrine approach originally reported by Jugé and co-workers in the early 90s, with sequential displacement of the chiral auxiliary by alkyl/aryl lithium reagents. Likewise, the conversion of the original phosphinous acid borane into other P-chiral derivatives uses standard reagents for these types of transformations. However, the fact that the chiral auxiliary in this case can be prepared on 50 gram scale does make this an interesting alternative to access P-chiral phosphines. I would suggest submission to a journal specializing in organic synthesis, such as *J. Org. Chem*, *Eur. J. Org. Chem.* or *Synthesis*, following revision suggested below.

Page 3, Scheme 3 Do the authors have any explanation as to why the expected reaction of 1e with RPOCl₃ does not take place?

Page 4, Scheme 5 Were any alkynyl Grignard reagents investigated in this study?

Can the ee be improved by recrystallization in those cases where the enantioselectivity was modest?

Page 6, Scheme 10 The enantiodivergent synthesis of the two different enantiomers of a chiral phosphine simply by switching the addition order of the lithium/Grignard reagents was described already by Jugé in 1990 (*Tetrahedron Lett.* 1990, 6357). The Jugé group should be credited for this observation in this context.

An example of the application of one or a few of the prepared compounds as ligands or organocatalysts in an asymmetric transformation would strengthen the manuscript.

In addition to reference 3a, there is a more recent review by Imamoto, giving an overview of synthetic routes to P-chirogenic phosphorous compounds, that could be cited (Proc. Jpn. Acad., Ser. B 2021, 97).

The Supporting Information is well prepared and the information provided is sufficient for other researchers to reproduce the results.

Dear reviewers,

Thanks very much for your positive evaluations and constructive suggestions to our manuscript titled "Universal and Divergent P-Stereogenic Building with Camphor-Derived 2,3-Diols (CAMDOL)". We have carefully checked your comments and revised our manuscript, taking into account all the points raised. All the changes in the revised manuscript text file have been highlighted. The following are our responses to your comments one by one.

Reviewer #1 (Remarks to the Author):

Chemistry of chiral organophosphorus compound is of utmost importance in the field of chemical sciences. In organic synthesis new ligands based on chiral phosphines are routinely used to facilitate challenging asymmetric transformations. Moreover, in the field of medicinal chemistry chiral phosphate functionalities play critical role in development of new oligonucleotide therapeutics. Therefore, development of new asymmetric routes to these molecules is crucial to sustain growing demand for novel chiral-at-phosphorus scaffolds. In recent years, numerous practical solutions to this problem were presented either based on chiral auxiliaries (ACS Cent. Sci. 2021, 1473) or catalytic approach (Nat. Chem. 2023, 714).

In this article authors introduce new type of camphorquinone-derived chiral auxiliary (CAMDOL), used for the stereocontrolled synthesis of chiral P(III) and P(V) compounds. The authors describe in detail optimization efforts leading to the development of the reagent and its later application in synthesis of diverse phosphinous acid boranes, as well as their further transformations leading to a range of P(III) and P(V)-based molecules. The products are obtained in good to excellent yields and with high stereocontrol, by treating the chiral-P(III) precursor **4c** with organometallic reagents (Grignard or Li-based). The major novelty of this work is the application of new chiral-pool-derived auxiliary to the synthesis of asymmetric P(III) reagents. In my opinion this work could be of use in development of new type of chiral phosphine ligands and phosphinous acid boranes. On the other hand, applications of this methodology to access more complex P(V) scaffolds (i.e. nucleotide isosteres) will likely be limited due to the use of highly reactive and moisture-sensitive P(III)-reagents (i.e. RPCl_2).

Overall, I believe that that this manuscript meets the level of novelty required for the publication in Communications Chemistry.

Bellow I enlisted some comments and corrections to be addressed by the authors before final publication:

1. The proposed intramolecular ring closing of epoxide (accompanied by elimination of the phosphinate) is unlikely to proceed via $\text{S}_{\text{N}}2$ -like mechanism as depicted on Fig.11 (Int_CAM). Formation of the epoxide on a 6-membered ring via $\text{S}_{\text{N}}2$ requires antiperiplanar orientation of

nucleophilic O and leaving group, which in case of **Int_CAM** is not possible. It is more likely that this step follows S_N1 mechanism, i.e. under mildly acidic conditions (NH₄Cl) C-OP bond undergoes heterolytic cleavage leading to phosphinate anion and stabilized tertiary benzylic carbocation, followed by the nucleophilic attack of hydroxyl group leading to epoxide.

Response: We have corrected the proposed reaction mechanism and stated in the manuscript as following: The organometal reagent prefers to attack the frontside C²-hydroxyl P-O bond rather than the backside C³-hydroxyl P-O bond probably due to the C¹-hydrogen bond chelating to form a kind of stabilized six-membered ring. Under the mild acidic conditions with NH₄Cl solution, the C³-OP bond undergoes a heterolytic cleavage leading to the formation of P-chirality retained phosphinous acid product and the stabilized tertiary benzylic carbocation **Int CAM**. The following intramolecular S_N1 nucleophilic attack of C²-hydroxyl group at C³-carbocation of **Int CAM** then leads to the formation of camphor-epoxide (Scheme 13).

2. Since one of the major points of the publication is synthesis of chiral phosphinous acid boranes, the authors should cite state-of-the-art publications describing strategies for the asymmetric synthesis of these molecules (e.g. menthol-auxiliary-based approach developed by Buono et al: Chem. Commun. 2008, 3031 doi: 10.1039/B802817F; JACS 2011, 10728 doi: 10.1021/ja2034816).

Response: We have described the state-of-the-art about the synthesis of chiral phosphinous acid boranes in the manuscript as following: More importantly, the products of P(III) phosphinous acid boranes after deprotection of BH₃, are able to tautomerise to the P(V) secondary phosphine oxides (SPOs) which are high valuable compounds with a huge potential in asymmetric synthesis, catalysis and coordination chemistry.²² The advantages of new-emerging P-chiral SPOs over the most often used phosphine ligands are threefold: air and moisture stability, supramolecular bidentate formation, and bifunctional ligands activity. However, only a few efficient examples are known so far. The most practical approach employed for SPO-BH₃ was a multi-step procedure starting from the diastereomerically enriched menthyl *H*-phosphinate precursors, which however normally need twice recrystallization manipulations at low temperature, unfortunately always in very low yields.²²⁻²³ More general and competitive enantioselective methodology towards SPOs still remain a very challenging topic now.

3. The authors state in the text that both enantiomers of the product can be obtained through selection of desired CAMDOL auxiliary (i.e. derived from (+)- or (-)-camphorquinone). However, all compounds presented in the paper are obtained from (-)-camphorquinone-derived CAMDOL. The authors should back up their statement by comparing yield and ee for enantiomers of one of the products derived from both enantiomers of CAMDOL.

Response: We have accomplished the enantiodivergent synthesis of compound **5bo** by using (+)- and (-)- camphorquinone enantiomeric forms and the comparing results are shown in Scheme 10.

4. In general, there are a lot of grammatical and stylistic errors throughout the text. Before publication I recommend that the manuscript should be partially rewritten with a help of native English speaker.

Response: We have corrected all the grammatical and stylistic errors and polished the whole article as best as we can.

5. Supporting Information: Multiplicity of P signal in ^{31}P NMR for multiple phosphinous acid boranes (compounds 4ce, 4be, 5ai, 5aj, 5bd, 5bh, 5bi, 5bj, 5bk, 5bl, 5bm, 5bn, 5bo, 5cb, 6a, 6b) is incorrectly assigned as doublet. ^{10}B nucleus has spin = $3/2$, which gives rise to 1:1:1:1 quartet on ^{31}P NMR. Apparent doublet is results from the signal broadening.

Response: We have corrected all of the above incorrect ^{31}P NMR assignments.

6. Supporting Information: ^{13}C NMR spectrum for compound **1b** is missing.

Response: We have attached the ^{13}C NMR spectrum of compound **1b**.

Reviewer #2 (Remarks to the Author):

The manuscript of Shi, Xiao and coworkers describes the use of readily available camphor-derived 2,3-diols (CAMDOLs) as chiral auxiliaries for the preparation of optically pure, P-stereogenic P(III) and P(V) derivatives, including phosphine-boranes, phosphinous acid-boranes, phosphinites, phosphine oxides and phosphinates, among others.

The manuscript describes a new family of chiral, C1-symmetric diols, derived from camphorquinone as heterobifunctional auxiliaries for the preparation of optically pure P-stereogenic compounds by successive substitutions at phosphorus. The key compound and the most studied is the dioxaphospholidine-borane 4ce.

Despite the long history and the activity in the field of P-stereogenic ligands in he last 25 years,

especially as ligands for enantioselective catalysis, there are still very few relatively versatile synthetic methodologies. Indeed, only the Jugé-Stephan method (based on ephedrine) and the Evans-Imamoto method (based on sparteine) are relatively general.

The idea behind the manuscript – using a heterobifunctional auxiliary for the preparation of P-stereogenic compounds by successive substitutions – has been known for a long time as Figure 1 depicts. In this figure, however, only the Jugé's ephedrine method is commonly used.

In this context, the use a novel cheap and available auxiliary and the the large number of compounds described in the paper is very interesting and for this reason I am quite supportive about the acceptance of the manuscript.

Depite this, I have some comments in order to improve the results presented in the manuscript.

1. The Jugé-Stephan method. The Jugé-Stephan Method (reference 5) is based on ephedrine but in several occasions in the paper (for example in the caption of Figure 1) it seems to imply that the method use any bifunctional auxiliary. This is not correct and has to be corrected to make it clear that the Jugé-Stephan methodology is based on ephedrine but that other methods are similar, but using other heterobifunctional chiral auxiliaries.

Response: We have changed the word of "Jugé-Stephan" into more suitable "Jugé-Stephan-like" or "Jugé-Stephan type" as necessary in the context.

2. Advantages of CAMDOLs. One of the disadvantages of the Jugé-Stephan methodology is that ephedrine is a key precursor of the psychotic substance methamphetamine and therefore is tightly controlled, this could be stressed in the paper as an additional advantage of the methodology presented.

Response: We have added the description in paragraph 2 as following: However, one of the disadvantages of the practical Jugé-Stephan methodology is that ephedrine is a key precursor of the psychotic substance methamphetamine and tightly controlled, thus not amenable for scale-up activities.

3. Phosphinous acid-boranes. The methodology described in the manuscript yields phosphinous-acid boranes, an understudied type of auxiliaries that is used to prepare phosphinite-boranes, phosphine-boranes, oxides and sulfides. This is perfectly fine but I think it could be improved in two ways:

o The conversion of phosphinous acid-boranes to phosphinite-boranes by O-methylation with trimethylsilyldiazomethane is not so common in the literature and according to the manuscript appears to be quite general. This reaction should be emphasized and referenced appropriately in the manuscript.

Response: Actually, we firstly examined the *O*-methylation with MeI but failed. However, the trimethylsilyldiazomethane approach reported by Ding, exhibited high efficiency. Considering that it's not the research emphasis, so we just proceeded the work with this method. Of course, we have added in the revised paper the related information and reference (Ref. 24).

o The deboronation of phosphinous acid-boranes presented in the manuscript would furnish Secondary Phosphine Oxides (SPOs), which are very interesting compounds since they tautomerise to the P(III) phosphinous-acid, which are competent ligands with a huge potential in catalysis. Therefore, the deprotection of a few phosphinous acid-boranes **5** should be described and references about SPOs (for instance Catal. Sci. Technol., 2019, 9, 5504-5561; ChemcatChem, 2020, 12, 3982-3994; Synthesis, 2022, 54, 271-280) added to the manuscript.

Response: According to this wonderful suggestion, we have examined the HBF₄-mediated deprotection of phosphinous acid-boranes and afforded the SPOs **12a-d** in 95-98% enantioselectivities, which are summarized in Scheme 12.

Overall, I support the publication to Communications Chemistry if the points raised are appropriately addressed.

Reviewer #3 (Remarks to the Author):

The manuscript by Shi and Xiao describes the synthesis of P-stereogenic P(III) and P(V) compounds using a camphor-based chiral auxiliary as a template for installing substituents at phosphorus. While I find the manuscript and Supporting Information well prepared, I do not find the novelty sufficient for a high impact journal such as Communications Chemistry. The methodology is similar to the ephedrine approach originally reported by Jugé and co-workers in the early 90s, with sequential displacement of the chiral auxiliary by alkyl/aryl lithium reagents. Likewise, the conversion of the original phosphinous acid borane into other P-chiral derivatives uses standard reagents for these types of transformations. However, the fact that the chiral auxiliary in this case can be prepared on 50 gram scale does make this an interesting alternative to access P-chiral phosphines. I would suggest submission to a journal specializing in organic synthesis, such as J. Org. Chem, Eur. J. Org. Chem. or Synthesis, following revision suggested below.

1. Page 3, Scheme 3 Do the authors have any explanation as to why the expected reaction of **1e** with POCl₃ does not take place?

Response: It's a very surprising and some puzzling result to us too. To our best knowledge, PCl₃ possesses more reactivity and less hinderance than POCl₃, which may be the main reason. The similar phenomenon was also reported with tertiary diols of TADDOL. So we added the explanation in the manuscript as following: Compared with POCl₃, PCl₃ is relatively more reactive and less bulky, which may contribute to the above distinguished behaviour when introducing P-atom onto the CAMDOL's tertiary hydroxyl groups, similar to that of TADDOL's reactions.²⁰ (Page 3).

2. Page 4, Scheme 5 Were any alkynyl Grignard reagents investigated in this study?

Response: Accordingly, we have examined the phenylalkynyl Grignard reagents and obtained the phosphinous acid borane **5ai** in 82% yield and 80% *ee* value. The result has been added in Scheme 5 and SI.

3. Can the *ee* be improved by recrystallization in those cases where the enantioselectivity was modest?

Response: Actually, there are only 5 products with *ee* values less than 90% in this work including compounds **5ai** (80% *ee*), **5aj** (74% *ee*), **5ak** (83% *ee*), **5bq** (82% *ee*), and **5br** (65% *ee*). Regretfully, compounds **5ai** (80% *ee*), **5bq** (82% *ee*), and **5br** (65% *ee*) are all in liquid forms, thus not able to be recrystallized. As for compounds **5aj** (74% *ee*) and **5ak** (83% *ee*), their higher enantioselective forms have been given in Scheme 6 as **5be** (99% *ee*) and **5bh** (98% *ee*).

4. Page 6, Scheme 10 The enantiodivergent synthesis of the two different enantiomers of a chiral phosphine simply by switching the addition order of the lithium/Grignard reagents was described already by Jugé in 1990 (Tetrahedron Lett. 1990, 6357). The Jugé group should be credited for this observation in this context.

Response: The Jugé group's work have been credited in the revised paper (Page 6, Scheme 11; Ref. 25).

5. An example of the application of one or a few of the prepared compounds as ligands or organocatalysts in an asymmetric transformation would strengthen the manuscript.

Response: It's a very good idea. Actually, we have commenced this work. We would like to present the results in another paper which we deem it to be beyond the scope of this article. Hope you would like to accept that.

6. In addition to reference 3a, there is a more recent review by Imamoto, giving an overview of synthetic routes to P-chirogenic phosphorous compounds, that could be cited (Proc. Jpn. Acad., Ser. B, 2021, 97).

Response: We have added the review paper in the ref. 3e.

The Supporting Information is well prepared and the information provided is sufficient for other researchers to reproduce the results.

REVIEWERS' COMMENTS:

Reviewer #2 (Remarks to the Author):

The authors have appropriately addressed the comments of the reviewers so I recommend publication of the manuscript.